# Determinants of a Firm's Sustainable Competitive Advantages: Focused on Korean Small Enterprises

**Sora Lee** [ID] **and Jaewon Yoo** *[ID]

Department of Entrepreneurship and Small Business, Soongsil University, Seoul 06978, Korea; sora.lee@ssu.ac.kr
* Correspondence: yjw1774@ssu.ac.kr

**Abstract:** This study identifies an integrated model of a firm's sustainable competitive advantages that helps understand how market orientation is related to an organization's sustainable competitive advantage. An empirical test of the proposed framework utilized data from 312 top management team members or project managers in Korea to access and evaluate resource input levels, organizational capabilities, and overall environmental contexts; it indicates that market and technological resource input and marketing and innovative capabilities mediate the positive impact of market orientation on a firm's competitive advantages. The findings suggested that technological turbulence's moderating role weakened the positive effect of technological resource input on innovative capability. Contrastingly, market turbulence has not moderated the influence of market resource input on marketing capability. The results call on management to understand the internal market orientation process to enhance the presence of environmental turbulences in industries, thus increasing the competitive advantage of firms.

**Keywords:** market orientation; market resource input; technological resource input; marketing capability; innovative capability; market turbulence; technological turbulence; sustainable competitive advantage; market performance; small enterprises



## 1. Introduction

Despite the significant advances in understanding the different performances through corporate resources, recent studies have focused more on the resource deployment process rather than resources input [1]. The benefit of utilizing market resources has been a salient topic in the marketing field in terms of sustainability [2–5]. By developing and exploiting resources, firms will achieve a superior position in the marketplace [6].

Former scholars have endeavored to integrate RBV (resource based view) of the enterprises and the dynamic capabilities based on the marketing theory [2] and defined market orientation as valuable, rare, socially complex, and causally ambiguous firm-level resources [7]. However, some researchers have expressed doubt on the stand-along role of market orientation as a source of competitive advantage. Because of this, the demand for integrated market orientation, with other dynamic capabilities, has been raised.

The research of market orientation, besides marketing and innovation capabilities, has been progressed separately in management literature [8]. Marketing capability focuses on creating customer demand and giving customers an invaluable proposition [9]. Meanwhile, innovation capability focuses on investigating new chances rather than merely exploiting retained strengths [7]. According to Ngo and O'Cass (2012), innovation and marketing capabilities are means for actualizing a firm's market orientation [10]. Firms are devoted to exploring capability, specifically innovation and marketing capabilities [10].

According to the RBV theory, a unique set of capabilities and resources is given to each firm, and some capabilities have more influential power on organizational outcome than others [11]. Such a difference in impact is attributed to the efficiency with which a firm can convert its resources to valuable (and difficult to imitate) capabilities, as well as

into performance. However, firm capabilities are more refined and invaluable through ceaseless investments over time [12]. The raises two questions: how dominant can firm investments be made? What is the role of investment in building valuable organizational capabilities to be sustainable in the market?

While some scholars argue that firm strategies are implemented through developing capabilities [13], little attention has been devoted to exploring resource input and capabilities as mechanisms through realizing a firm's market orientation. Despite the important role of resource investment in building valuable capabilities, as well as a sustainable comparative advantage, most work has heavily focused on resource and capability relationships. There is a dearth of empirical research on the effect of resource investment on the nexus between resource and capability. Furthermore, a review of the RBV and capability theory demonstrates that the relationship between resource input (marketing and technical) and capability (marketing and innovation) is in the black box. The demand to unveil the mechanisms of how these firm resource inputs and capabilities mediate market orientation and performance has been raised.

Addressing the mediating role of resource input in the RBV framework will be beneficial to researchers and managers by increasing our understanding on how to manage organizational resource input to enhance specific capabilities, especially for small and medium-sized enterprises (SMEs). Furthermore, we consider the environment turbulence as a moderated variable to identify the contingencies in the relationship between specific resource input and capability.

According to the Korean government, the number of SMEs in Korea is almost 6.3 million, accounting for 99.9% of the total industry in 2017. The Korean economy experienced dramatic growth, also known as "Miracle of Han river", from 1960 to the 1970s, thanks to technological innovation, but it now faces severe polarization between large firms and SMEs [14]. It is important for the Korean government to understand the resources and capabilities of SMEs. However, recognizing partial things is not enough, it is necessary to know the integrated model for sustainable growth and performance.

Thus, our primary aim is to investigate the relationship between marketing orientation and capabilities (marketing and innovation) and identify the mediating role of resource inputs in the relationship between resource (market orientation) and capabilities (marketing and innovation). The second research objective is to develop and empirically identify the contingency effect (marketing and technical turbulence) in the relationship between resource input and capabilities. The final objective is to investigate how each capability affects organizational sustainable advantage, resulting in increased market performance. Therefore, we can have a profound understanding of the determinants of sustainable advantages for small companies.

The rest of the paper is structured as follows. In the second section, we offer the theoretical underpinning of using RBV framework and the conceptualization of market orientation of the firm, followed by a discussion of research methods and results in Sections 3 and 4. Section 5 concludes.

## 2. Literature Review and Hypotheses

### 2.1. Resource-Based View and Market Orientation

As a salient frame of suggesting performance differentials among firms, RBV has been investigated between marketing scholars [15]. It encourages firms to explore resources because diverse resources are a fundamental ground for the differentiated and superior advantages in the marketplace [15]. Barney (1991) clarifies resources as a bunch of capabilities, attributes, and assets, in addition to organizational processes, information, and knowledge [15]. Amit and Schoemaker (1993) suggested that firm resources are categorized by two components: tangible (physical capital and financial assets) and intangible (human capital and technical know-how) [16]. Meanwhile, capability is described as the process of optimizing a firm's given resources to achieve the desired end [16]. Capabilities, which are



considered as invisible assets and all types of organizational processes, are formed by a firm over time that cannot easily be gained [17].

RBV researchers argue that a firm will obtain differential resources and diverse levels of capabilities. Firms' competitive advantages are achieved by creating new resources, facilitating capabilities platforms, and making inimitable capabilities. However, not all resources are regarded as important in creating competitive advantage. Advantage-generating resources are those that possess the combined traits of enabling the provision of competitively outstanding value to customers [15], un-duplicable by competitors [18], and whose value can be fitted by the organization [19].

Despite indisputable strength of resources and performance differentials, there have been controversial views on elucidating how resources are arranged to attain predominant firm outcomes. Amit and Schoemaker (1993) propose that resources acquire value in a precisely targeted market [16]. Collis and Montgomery (1995) also suggest that the casual relationship between a valuable resource and satisfaction of customer needs is significant and indisputable. The adoption of the RBV framework to analyze organizational performance has been encouraged [20,21] and deep comprehension between marketing and other functional capabilities and their impact on outcomes have been researched [11,22,23]. Many different resources, which underpin marketing activities, generate significant competitiveness. The representative examples, including brand image, customer relationship, and marketing orientation, have some distinctive traits. Typically, they are hard to form, take a long time to build, rely on tacit knowledge and skills, and are socially complex.

Based on strategic management and marketing, authors add market orientation to incorporate market resources that have value in the marketplace for SMEs. Market orientation, organizational-level values and beliefs, make firms put customer needs at the top of the business plan. Varela, Gutierrez, and Antón (1998) distinguish between market orientation as a philosophy and as a cultural tradition [24]. However, Varela et al. (1998) compartmentalize the term as a culture when defining these perspectives [24].

Another approach views market orientation as a strategic or behavioral approach, which characterizes the processing of market information and inter-functional coordination of market information. Since ways of thinking delimit ways of acting, these approaches have been frequently defended.

Besides these two approaches, several researchers regard market orientation and the capacity of a company to provide a sustainable competitive advantage as resources [3,4,24,25]. Thus, it is possible to perceive market orientation from the theory of resource and capacities perspective. Armario (1995) also identified that such a theory can account for organizational competitiveness by regarding market orientation as an internal factor within an organization and explain its competitiveness [26].

A firm's market orientation is considered a major distinguishing resource closely related to overall performance [27,28]. Orientation is complex and takes time to build; it is based on experience and tacit skills and is hard to convey from one organization to another. For example, if a manager in one company was headhunted by a different company, or a competitor from a different embedded orientation, his or her capabilities may be less effective. Market orientation is the basic organizational resource that supports organizational processes spanning the boundaries of marketing. The applicable range of orientation includes management of human capital, operation processes, product, and service innovation, and even financial management [29]. RBV postulates that firms obtain economies of scale by diversifying their resources and capabilities and sharing their fixed assets, such as distribution channels, common production facilities, or even brand name [30].

### 2.2. Research Model Development

As a valuable, uncommon, socially complex, and casually ambiguous resource available to form, market orientation allows firms to create and sustain competitive advantage [4]. Hooley et al. (2005) also suggest that market orientation is considered a deeply



embedded culture in the firm, and is able to form a distinctive resource [29]. As the focus of market orientation is on the customer, a firm with market orientation listens to customers and suggests solutions based on the needs and wants of the customers [31,32]. Market orientation also make firms concentrate on competitors' moves to keep and enhance their market positions over competitors [33,34]. Market-oriented firms follow specific and identifiable routines and processes, such as generating information throughout the firm and modifying business strategies to improve customer value [27,35].

The weakness of market orientation as a standalone source in creating a competitive edge fosters the demand for additional complementary resources to enhance its competitive value [36]. Therefore, without considering certain capabilities, the unique strength of market orientation can be diminished [7,37].

Since a market-oriented firm fully concentrates on the expressed and even unexpressed desires of its clients [38,39], it should acquire talents to realize customers' wishes and needs and to find unique solutions to unexpressed desires. Marketing capability can be defined as a firm's interrelated organizational regular things for carrying out marketing activities, such as 4Ps (product, pricing, place, and promotion), planning marketing strategies, and marketing implementation [22,37]. Firms pursuing a strong market orientation are more likely to develop a higher order marketing capability to achieve customer-related advantages concerning attracting and satisfying customers, building strong relationship with customers, and, finally, retaining customers. A high level of market orientation encourages a firm's entire internal functions to be involved in their role and contribution to offer superb value to customer [29]. Increased market orientation leads to heightened levels of market innovation [32] and the development of superior customer linking capabilities [3]. Ngo and O'Cass (2012) also found the positive effect of market orientation on marketing capabilities [10].

However, both Day and Wensley (1988) and Hunt and Morgan (1995) suggested that a firm's competitive advantage will prolong as long as it invests in capabilities and innovation, which compel competitors into catch-up mode [4,33]. Ngo and O'Cass (2012) suggest that a firm's capabilities, which originate from market orientation, could be more refined and valuable through constant investment as time goes on [10]. Based on this research, we suggest that the relationship between market orientation and marketing capabilities will be mediated by marketing resource input.

**Hypothesis 1:** *Marketing resource input mediates the relationship between market orientation and marketing capability.*

Innovation capability is described as a firm's ability to create and offer solutions to solve customers' concealed or unconcealed problems [40]. Hurley and Hult (1998) assert that the innovation ability makes a great contribution to a firm's competitiveness and expands the area of product development, operation process, and marketing [41,42]. Innovation capability is described as organizational tasks to perform innovation activities and is closely related to a firm's products and services, production process, management, market, and marketing [22,37].

Several studies indicate that market orientation plays a part in corporate development —of innovation capability [41,43]. Deshpandé, Farley, and Webster (1993) assume that market orientation has a causal relationship with innovation and Han et al. (1998), and provide the empirical proof of linkage between the cultural market orientation and innovation [31,42]. Mavondo, Chimhanzi, and Stewart (2005) also suggest that market orientation has a positive connection with three types of innovation: process, product, and administrative innovation [44]. The meta-analysis of Kirca, Jayachadran, and Bearden (2005) argues that market orientation has an impact on performance through innovation [45].

However, investment in capability preceded by building capability and continual investment based on the market orientation will lead to innovation capability [4,34]. Therefore, market orientation will increase the level of technical resource input and then enhance organizational innovation capability.

**Hypothesis 2:** *Technical resource input mediates the relationship between market orientation and innovation capability.*

Former researches have recognized that potentially outer environmental factors can have an influence on the extent of market orientation on corporate outcomes [38,46]. For SMEs, technological and social upheaval are factors that have a major impact on firms [47]. Specifically, fluctuations in market and technology have been typically derived from diversified consumer preferences or variability of industry technological standards, respectively. Market turbulence can be defined as the rate and predictability of change of the customer segments; their reference and technological turbulence are referred to as the degree and predictability of change engaged in product and process technologies in an industry [48].

Market turbulence, on the one hand, increases the level of ambiguity and risk of business value chain procedures, and on the other hand, raises the degree of causality between strategy and firm performance [49]. Therefore, it is a critical factor in building marketing capabilities. A turbulent market exhibits characteristics of frequent and unpredictable changes in production preferences and customer needs in product and marketing activities. Many researchers suggested that the effect of market orientation may decrease when the market goes through severe change. Several empirical results have discovered the dark side of market orientation in the rapid changing situation [50–52].

In this research, we suggest that the effect of marketing resource input on marketing capability will be diminished when the market faces rapid changes. In an unpredictable market, predicting customers' needs is not easy, and reacting to the change through development of new products may not be swift enough, and implementing accurate resource investment based on the understanding of immediate change in customer needs is very difficult to execute. Therefore, the positive effect of marketing resource input on marketing capability will be diminished when the market is highly turbulent.

**Hypothesis 3:** *Market turbulence negatively moderates the relationship between marketing resource input and a firm's marketing capability.*

Firms with high technological variation experience higher levels of change in production and technological processes than those with low technological variation [53,54]. Because of the rapid technological change, firms find it hard to catch up with changing technological demands to stay in circumstances with high technological turbulence.

A highly technologically turbulent condition is described by a short cycle of technological innovation and obsolescence. In a dynamic business environment, the timely introduction of new products to replace obsolete products may burden firms due to lack of resources. Therefore, under high technological turbulence, firms will not be able to easily determine what technology areas they should invest in. Hanvanich et al., (2006) argued that radical technological changes restrain corporations from adopting innovation and prevent them from actively looking for appropriate technological investment sectors [48]. Consequently, we believe that the effect of technological resource input on innovation capability will decrease when technological turbulence is high.

**Hypothesis 4:** *Technological turbulence negatively moderates the relationship between technical resource input and a firm's innovation capability.*

According to RBV, competitive advantage can be achieved by diversifying and expanding a firm's resources into a different market and businesses. Capabilities, which are more firm-specific factors and less transferable than resources, make firms produce more superior business outcomes. Considering intensive market competition and shortening of product life cycles, firm capabilities to develop innovative marketing offerings has attracted the attention of scholars and practitioners [10]. Thus, a firm's competitive advantage depends on innovation capabilities of products or services.

Some scholars have defined marketing capability as an integrated procedure in which a firm utilizes its visible and invisible resources to comprehend a consumer's precise needs,

launch differentiated products relative to the competition, and acquire predominant brand equity [3,11,20]. A firm's marketing capabilities is developed by combining employee skills and knowledge to the available resources [55]. A firm that invests more resources to build and maintain a relationship with its customers can improve market-sensing capabilities [56]. Such marketing capabilities are hard to imitate and utilize for rival companies [3]. Several empirical studies also support the positive connection between marketing capability and sustainability [57,58]. Na, Kang and Jeong (2019) investigated how marketing innovation and the kind of marketing activities, affect sustainable competitive advantage. Kamboj and Rahman (2017) also tested the relationship between marketing capabilities and competitive advantage and found a positive connection. Thus, marketing capability is regarded as a crucial source to improve the competitive advantage of firms.

**Hypothesis 5:** *Marketing capability is positively related to a firm's competitive sustainable advantage.*

The definition of innovation capability is based on customer's current and future needs [40]. According to Hurley and Hult (1998), the innovation capability generates a firm's competitiveness and spans across value chain to meet customer needs [41,42]. Besides adopting an organizational innovative culture, a firm can attract concrete, direct investment in developing its innovation capability through investment in research and development (R&D) or strategic technology alliance. Superior innovation capability indicators, such as patents, add to a firm's large-scale investment in R&D, and increase investor intention (to invest in the firm) and improve the quality of innovations. Moreover, innovation capabilities, such as technological innovation, are positively related to small family firm performances [59].

It is stated that technological and innovative capabilities of a firm can be a fountainhead of competitive advantages and exceed general performance, according to the RBV and knowledge-based view [60]. Other scholars have tried to investigate the relationship between capabilities and firm sustainability [59,61,62]. Lai, Lin, and Wang (2015), as well as Kamboj and Rahman (2017) found a significant positive relationship between innovation capability and sustainability. Moreover, Ngo and O'Cass (2012) suggest that innovation capability proposes clever ideas for change and improvement, and may result in product and process innovation advantages [10]. Therefore, the following hypothesis is presented.

**Hypothesis 6:** *Innovation capability is positively related to a firm's competitive sustainable advantage.*

Competitive advantage refers to the extent to which an organization can assume a defensible position over its rivals [63]. According to Barney (1991), competitive advantage is generally conceptualized as the implementation of a strategy not currently being implemented by other firms, which facilitate the reduction of costs, the exploitation of market opportunities, and the neutralization of competitive threat [15]. Market performance refers to the firm's ability to satisfy and retain customers by offering quality products and services [64].

Though the term competitive advantage and performance are often used interchangeably, these two constructs are recognized to be conceptually distinct [65]. Having a competitive advantage generally suggests that an organization can have one or more of the capabilities, such as lower price, higher quality, higher dependability, and shorter delivery time when compared to its competitors. These capabilities can lead to high levels of economic performance. Because such benefits tend to enhance customer loyalty and perceived quality [66], a firm that can exploit its resource-capability combinations to effectively attain a differentiation-based competitive advantage can improve its performance by selling more units at the same margin or be selling the same number of units at a greater margin [67]. Existing research has well-documented the positive effect of competitive advantage on performance, as competitive advantage provides a firm with the wherewithal to outperform its competitors [68,69]. Newbert (2008) also identifies the positive effect of competitive advantage on market performance [67]. (Figure 1).

**Hypothesis 7:** *Firm's competitive advantage is positively related to market performance.*

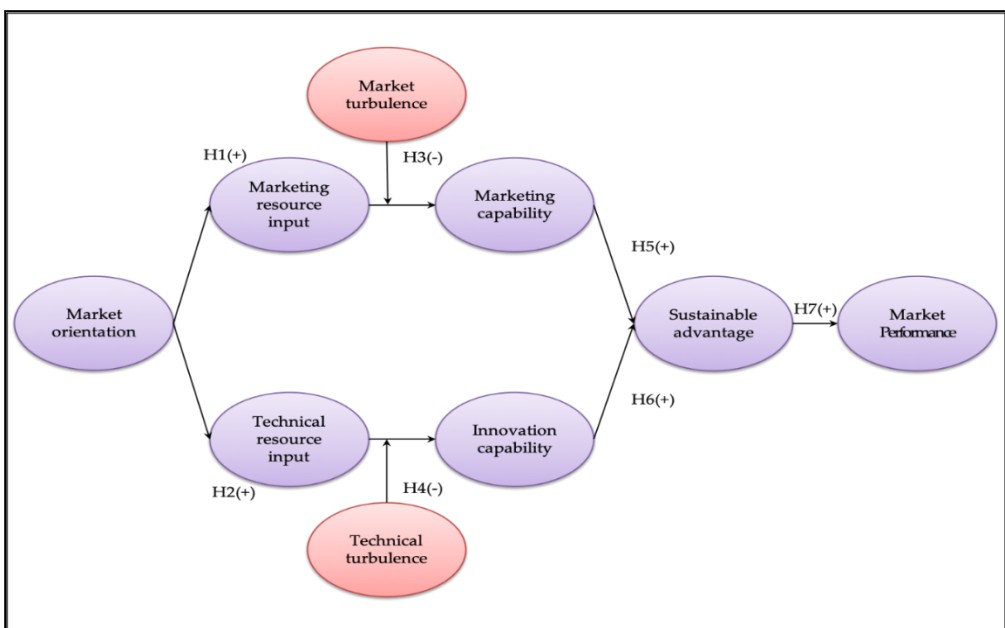

**Figure 1.** Research model.

## 3. Research Methods

### 3.1. Measures

The item indicators were measured by a seven-point Likert-type scale with anchors of "strongly disagree" and "strongly agree." Market orientation was measured using four items from Nguyen et al. (2016) conceptualized by Narver and Slater (1990) [27,70]. It contains competitor, sharing information and customer-oriented behaviors, which were tested by former researchers [71,72].

Resource inputs (marketing and technological) were measured with the four-team scale developed by Cui et al. (2013) respectively [73]. Marketing resource input consisted of four items to evaluate the degree of the resource input of an organization on environmental scanning, market planning, marketing implementation, and analyzing customer needs. Technological resource input was also measured with the scale, which rates the degree of resource input of the organization on search and development, industrial design, engineering management, and information technology [73].

Market and technological turbulence were adapted from Chen et al.'s (2012) study [74]. Market turbulence was measured by four items, reflecting the perceived speed of changes in customer needs and competitor actions. Technological turbulence was also measured with the four-item scale, representing the perceived speed of the variation and instability of the technological environment.

To measure marketing capability, we used the eight-item scales, consisting of two sub-dimensions (marketing planning capability and marketing implementation capability) developed by Chang, Park, and Chaiy (2010) [75]. Marketing planning capability was measured with four items and marketing implementation capability was also measured with four items.

Innovative capability is also a multi-dimensional construct, which consist of two sub-dimensions (radical innovative capability and incremental innovative capability). We measured each of these innovative capabilities with three items, per se, based on Subramaniam and Youndt 's (2005) study [76]. These higher-order constructs, marketing capability and technological capability, were summated and analyzed at the second-order factor level to assess the hypothesized links among the constructs. Therefore, we analyzed each two-item for marketing capability and innovative capability after checking the reliability of each construct with all items (four items).

The study measured sustainable competitive advantage using seven statements from Storey and Kahn (2010) [77]. The respondents indicated the extent to which they agreed or disagreed with all measures on the combination of a range of outcomes, such as the development of new markets for the better utilization of existing resources that give long-term benefits.

To measure market performance, we used the four-item scale, as suggested by Grawe, Chen, and Daugherty (2009) [78]. This scale asks subjects to indicate the firm's financial and market performance in the last year relative to major competitors. Detailed items are presented in Appendix A.

### 3.2. Sampling Procedure and Data Collection

The instrument was initially prepared in English and then translated into Korean. For this study, the data were collected from small business entrepreneurs or those who were in charge of the decision making of the firm. The firm (or cooperative) president who received, or who was interested in receiving government support (funding, education, consulting, etc.), was invited to participate in the annual conferences for free. At the conference, the annual achievements were shared, and plans for the next year were explained. It took place in seven regions in Korea and the event schedule was different for each region. Due to the lack of space, only companies who expressed their intention to participate were listed on a first-come, first-served basis, which means a non-random sampling procedure. We carried out the surveys after the conferences, over three months, from October to December 2016. We made it clear concerning the purpose of the study, asked firm representatives to fill out questionnaires, and handed out pens (and the questionnaires) face-to-face. Autonomous responses were made, and pens were given as a gift when the survey was over. After eliminating surveys of respondents who did not answer most of the questions, or who checked the same survey numbers, we retained 312 useable surveys out of 409 (about 76% response rate). The majority of responses came from the metropolitan area (about 76%). Of the respondents, 49.7% were female, 44.9% of them had undergraduate degrees, and 12.8% indicated they had graduate degrees. On average, they had 12.39 years of business management experience and 38.8% of the respondents engaged in their company for more than 10 years.

## 4. Results

### 4.1. Measurement Model Analysis and CFA Results

The result of correlation analysis is shown in Table 1. Measurement assessment was conducted via confirmatory factor analysis (CFA). After deleting poorly loading items, the finalized fit indices were $\chi^2$ = 997.93 ($p$ = 0.00), df = 398, standardized root mean square residual (SRMR) = 0.056, root mean square error of approximation (RMSEA) = 0.058, comparative fit index (CFI) = 0.97, goodness of fit index (GFI) = 0.87, non-normed fit index (NNFI) = 0.98 and normed fit index (NFI) = 0.97. Based on Kline's recommendation, we presented crucial indices in the note below Table 2 [79]. Across our measurement models, factor and item loadings all exceeded 0.64, with all t-values greater than 14.02, providing evidence of convergent validity among our measures. As shown in Table 1, we tested the average variance extracted (AVE) of each latent variable and found that the values range from 67 to 79 percent, showing discriminant validity among variables [80]. Moreover, all measures exhibit strong reliability with composite reliabilities ranging from 0.86 to 0.96. Overall, constructs that we suggest demonstrate good measurement properties.

**Table 1.** Correlation analysis.

| Constructs | 1 | 2 | 3 | 4 | 5 | 6 | 7 | 8 | 9 |
|---|---|---|---|---|---|---|---|---|---|
| 1. Market Orientation | (0.67) | | | | | | | | |
| 2. Market Resource Input | 0.45 ** | (0.78) | | | | | | | |
| 3. Tech Resource Input | 0.37 ** | 0.56 ** | (0.79) | | | | | | |
| 4. Market Capability | 0.52 ** | 0.42 ** | 0.43 ** | (0.76) | | | | | |
| 5. Innovation Capability | 0.50 ** | 0.46 ** | 0.43 ** | 0.46 ** | (0.77) | | | | |
| 6. Market Turbulence | 0.29 ** | 0.26 ** | 0.32 ** | 0.2 ** | 0.30 ** | (0.70) | | | |
| 7. Tech Turbulence | 0.32 ** | 0.16 ** | 0.24 ** | 0.29 ** | 0.24 ** | 0.58 ** | (0.67) | | |
| 8. Sustainable Competitive Advantage | 0.51 ** | 0.49 ** | 0.45 ** | 0.47 ** | 0.52 ** | 0.31 ** | 0.18 ** | (0.76) | |
| 9. Market Performance | 0.41 ** | 0.45 ** | 0.45 ** | 0.38 ** | 0.36 ** | 0.20 ** | 0.18 ** | 0.56 ** | (0.84) |

Note. The numbers in diagonal parentheses indicate the average variance extracted (AVE) value, ** $p < 0.01$.

**Table 2.** Measurement model constructs.

| Constructs | Loading | C/R | Mean | Stdev | Skewness | Kurtosis |
|---|---|---|---|---|---|---|
| Market orientation | 0.64 ~ 0.90 | 0.85 | 5.38 | 1.16 | −0.72 | 0.37 |
| Market resource input | 0.81 ~ 0.93 | 0.91 | 4.56 | 1.2 | −0.16 | 0.13 |
| Tech resource input | 0.87 ~ 0.93 | 0.91 | 4.73 | 1.27 | −0.39 | −0.02 |
| Marketing capability | 0.80 ~ 0.94 | 0.86 | 5.15 | 1.16 | −0.49 | 0.11 |
| Innovation capability | 0.87 ~ 0.89. | 0.87 | 5.49 | 1.06 | −0.84 | 0.92 |
| Market Turbulence | 0.67 ~ 0.91 | 0.90 | 4.58 | 1.31 | −0.28 | −0.06 |
| Technological Turbulence | 0.73 ~ 0.90 | 0.89 | 4.31 | 1.29 | −0.1 | −0.27 |
| Sustainable Competitive Advantage | 0.83 ~ 0.91 | 0.96 | 5.42 | 1.09 | −0.85 | 1.18 |
| Market Performance | 0.87 ~ 0.95 | 0.95 | 4.73 | 1.3 | −0.48 | 0.13 |

Note. $\chi^2$ = 997.93 ($p$ = 0.00), df = 398, SRMR = 0.056, RMSEA = 0.058, CFI = 0.98, GFI = 0.87, NNFI = 0.98, NFI = 0.97

## 4.2. Results of Hypotheses Testing

Hypotheses tests were conducted based on two approaches. First, we tested our hypotheses using the PROCESS macro model, which enables us to check the mediating and moderating effect through bootstrap based on ML (maximum likelihood) [81]. The direct and indirect effects are reported in Table 3. As the table shows, the mediation effect of market resource input was positive and significant (β = 0.71, standard error(s.e) = 0.03), with Hypothesis 1 being supported. The indirect effect of market orientation on innovation capability was significant. Significant tests were based on bias-corrected confidence intervals by 5000 bootstrapped samples [82]. Thus, technological resource input mediated the relationship between market orientation and innovation capability.

**Table 3.** Results of mediated model.

| Category | Effect | Relationship | Est. | Boot s.e | LLCI | HLCL |
|---|---|---|---|---|---|---|
| *H1* | Direct | MO → MC | 0.35 | 0.04 | 0.41 | 0.56 |
| | Indirect | MO → → MRI → MC | 0.13 | 0.03 | 0.08 | 0.19 |
| *H2* | Direct | MO → IC | 0.71 | 0.03 | 0.65 | 0.77 |
| | Indirect | MO → TRI → IC | 0.09 | 0.02 | 0.05 | 0.13 |

Note. MO = market orientation, MC = market capability, MRI = market resource input, IC = innovation capability, TRI = technological resource input.

Next, we tested moderation effects utilizing process macro model 14 and the results are presented in Table 4. It shows that the negative and significant relationship (β = −0.060, s.e. = 0.020, $p < 0.01$) between technological resource input and technological turbulence, supporting Hypothesis 4. However, the interaction effect of market turbulence with market resource input was insignificant, which leads to rejection of Hypothesis 3. In addition, the variance inflation factors (VIFs) of predictor variables are less than 2, which suggests that a multicollinearity problem is not present in the applicable model (see Table A2).

**Table 4.** Results of moderated mediation effect.

| Dependent Variable | MC | | | | IC | | | |
|---|---|---|---|---|---|---|---|---|
| | *estimate* | s.e | LLCI | ULCI | *estimate* | s.e | LLCI | ULCI |
| Constant | 1.3221 ** | 0.175 | 0.978 | 1.666 | 3.687 ** | 0.214 | 3.267 | 4.107 |
| MO | 0.708 ** | 0.032 | 0.646 | 0.771 | 0.339 ** | 0.039 | 0.262 | 0.415 |
| MRI | 0.189 ** | 0.031 | 0.127 | 0.250 | | | | |
| MU | 0.022 | 0.027 | −0.031 | 0.075 | | | | |
| MRI X MU (H3) | 0.026 | 0.017 | −0.007 | 0.062 | | | | |
| TRI | | | | | 0.304 ** | 0.035 | 0.236 | 0.373 |
| TU | | | | | 0.042 | 0.033 | −0.024 | 0.107 |
| TRI X TU (H4) | | | | | −0.060 ** | 0.020 | −0.100 | −0.020 |
| *F* | | 211.765 | | | | 71.744 | | |
| $R^2$ | | 0.677 | | | | 0.150 | | |

Note. MU = market turbulence, TU = technology turbulence, s.e = standard error, LLCI = lower limit confidence interval, ULCI = upper limit confidence interval, ** $p < 0.01$. * $p < 0.05$.

Second, we used structural equation models (SEM) to test each relationship between independent and dependent variable. Table 5 presents the results of SEM, showing whether each path is supported or not. Reported model fit statistics were $\chi2 = 1338.47$, df = 223, RMSEA = 0.10, CFI = 0.96, NFI = 0.95, and GFI = 0.79. Market capability and innovation capability had a positive relationship with a competitive sustainable advantage, respectively. The effect of innovation capability ($\beta = 0.71$, $p < 0.01$) was stronger than market capability ($\beta = 0.29$, $p < 0.01$). As expected, competitive sustainable advantage has a positive effect on market performance ($\beta = 0.56$, $p < 0.01$). Therefore, hypotheses 5, 6, and 7 were supported.

**Table 5.** Results of the structural equation model.

| Category | Relationship | Est. | s.e | *t* |
|---|---|---|---|---|
| *H5* | Marketing capability $\rightarrow$ Competitive Sustainable advantage | 0.29 ** | 0.04 | 7.12 |
| *H6* | Innovation capability $\rightarrow$ Competitive sustainable advantage | 0.71 ** | 0.05 | 14.35 |
| *H7* | Competitive Sustainable advantage $\rightarrow$ Market performance | 0.56 ** | 0.05 | 11.24 |

**Note**. $\chi2$ (df) = 1338.47(223), RMSEA = 0.10, CFI = 0.96, NFI = 0.95, GFI = 0.79, ** $p < 0.01$.

## 5. Discussion

### 5.1. General Results Discussion

We believe that this study provides several contributions that enhance managerial and theoretical understanding of sustainability of small enterprises through integrated marketing models.

Generally, research on RBV suggest that market orientation enhances organizational capabilities, though the intervening mechanism of resource input in the resource-capability links has not been focused on. We found the mediating role of marketing resource input in the relationship between market orientation and marketing capability and technological resource input in the connection between market orientation and innovation capability in the small firm. Marketing and technological resource inputs are a concrete mechanism to reconfigure organizational resources for greater market and innovation capabilities through market orientation. Highly market-oriented and highly motivated firms tend to invest more resources in marketing and technology, which are better able to build a more qualified capability and have a greater chance of success.

Moreover, situational factors, such as resource input, show significant moderating effects on the development of organizational capability. When the market is turbulent, the relative effect of marketing resource input on building marketing capability is not significant. Because change in market is related to the heterogeneity of customer preferences and the rate for preference change [42], it is especially surprising that the impact of market resource inputs on marketing capability was not significantly moderated by market turbulence effect. One explanation for such a finding is that market resource is a fixed part

of organizational resource inputs because it is difficult to predict changes in the market environment in advance. Therefore, it is expected that organizations will always maintain a certain portion of market resource input ration regardless of market environment change.

In line with the contingency theory, the moderated effect of technological turbulence was adopted in our study. The positive effect of technological input on innovation capability is reduced when technology is highly turbulent. Thus, R&D investment in the more technologically turbulent industry may be harder to deal with; the effect of technological resource input on innovation capability is reduced. Therefore, firms confronting technological turbulent environments may be cautioned to decide on technological resource input to maintain innovation capability.

In this study, the mediating role of sustainable competitive advantage in the marketing and innovation capability and market performance was examined. Supporting the RBV argument—that the use of resources bring sustainable competitive advantage—is key to competitive advantage, which is why dynamic capabilities are essential. This study unboxes the internal procedure of the market orientation-resource input-capability-sustainable competitive advantage-performance relationship, in which the organization takes tactical measures to concentrate on market orientation, increase resource inputs, develop capability, and then create a sustainable competitiveness to improve market performance. Thus, identifying these sequential relationships serve a deep understanding of how market orientation convincingly has an influence on organizational market performance.

*5.2. Managerial Implications*

We also offer interesting practical implications based on our findings. While managers are encouraged to be market-oriented, the findings of this study highlight the significance of giving more managerial heed to the process, whereby market orientation affects market performance. Market orientation plays a role in enhancing organizational resource input in marketing and technology aspects and organizational capabilities. Market orientation itself may not be sufficient for firms to achieve expected performance, unless they transform market orientation into various kinds of resource investments and then into different capabilities. Thus, managers should understand the fundamental managerial process to perceive the latent value of market orientation. Based on understanding the internal process of market orientation, supervisors can identify the comprehensive relationship and concentrate on their efforts on enhancing key resource inputs and developing marketing and innovation compatibilities, enhancing market performance through competitive advantage.

Managers who take pains to align their firms' innovation capabilities with an unstable technological environment should figure out that, the careful control of technological resource inputs to handle environmental conditions is vital for achieving capability alignment, ultimately cultivating competitive advantage. As the cost-effectiveness of technological resource input decreases when technological environment is turbulent, the level of innovation capability decreases. Therefore, technological resource inputs, based on careful exploration and investigation of technological environment, enable them to efficiently manage their resources and maintain stable technological capability.

*5.3. Limitation and Future Research*

Our research needs to be qualified by the following considerations. First, the critical role of market orientation in the market and technological resource inputs did not include other strategic resource inputs and types of capability. Future studies should take other sorts of firm resource investments into consideration to realize the capability development process more thoroughly.

Second, the findings of this study should be read in the context of the limited sample size and the specific Korean cultural environment. Therefore, the results may not be generalized to the SME context in South Korea, but are probably useful as a qualified exploratory approach to analyze the relationship among the constructs.

Third, this study is limited by its cross-sectional design. The measurement method of the survey was based on participant experience. Although the methodology is well established, participants answering the questions based their responses on their own subjective evaluations, which may yield biased results. Future research should attempt to conduct a longitudinal study where the differences in relationships can be studied at various points in time, providing greater support for causality.

Finally, further understanding of the parsimony principle is needed. In consideration of the rejected moderating effect, alternative models may be proposed in future studies. In particular, some scholars suggest three types of turbulence: market turbulence, competitive turbulence, and technological turbulence. Future research may adopt other types of turbulence or try our model in other countries or situations.

**Author Contributions:** All authors contributed substantially to all aspects of this article. All authors have read and agreed to the published version of the manuscript.

**Funding:** This research received no external funding.

**Institutional Review Board Statement:** Not applicable.

**Informed Consent Statement:** Not applicable.

**Data Availability Statement:** The data presented in this study are available on request from the corresponding author. The data are not publicly available due to privacy.

**Conflicts of Interest:** The authors declare no conflict of interest.

## Appendix A

**Table A1.** Measurement items.

| Constructs | Item Descriptions | |
|---|---|---|
| Market orientation | 1. I frequently collect information on our competitors to help direct our marketing plans.<br>2. Market information is shared with all functions.<br>3. My firm's strategies are driven by the need to create customer value.<br>4. I seek to anticipate future customer needs. | |
| Market resource input | 1. The degree of environmental scanning of your firm.<br>2. The degree of market planning of your firm.<br>3. The degree of marketing implementation of your firm.<br>4. The degree of analyzing customer needs of your firm. | |
| Tech resource input | 1. The degree of research and development of your firm.<br>2. The degree of industrial design of your firm.<br>3. The degree of engineering management of your firm.<br>4. The degree of information technology of your firm. | |
| Marketing capability | Marketing Planning Capability<br>1. Super marketing planning skills.<br>2. Sets clear marketing goals.<br>3. Develops creative marketing strategies.<br>4. Thorough marketing planning process. | Marketing Implementation Capability<br>1. Allocating marketing resources effectively.<br>2. Delivering marketing programs effectively.<br>3. Translating strategies into action effectively.<br>4. Executing marketing strategies quickly. |
| Innovation capability | Incremental Innovative Capability<br>1. Innovations that reinforce your prevailing product/service lines.<br>2. Innovations that reinforce your existing expertise in prevailing products/services.<br>3. Innovations that reinforce how you currently compete. | Radical Innovative Capability<br>1. Innovations that make your prevailing product/service lines obsolete.<br>2. Innovations that fundamentally change your prevailing products/services.<br>3. Innovations that make your existing expertise in prevailing products/services obsolete. |

**Table A1.** *Cont.*

| Constructs | Item Descriptions |
|---|---|
| Market Turbulence | 1. Customer needs and product preferences change quite rapidly.<br>2. Customer product demands and preferences are highly uncertain.<br>3. It is difficult to predict changes in customer needs and preferences.<br>4. Market competitive conditions are highly unpredictable. |
| Technological Turbulence | 1. It is very difficult to forecast technological developments in our industry.<br>2. Technology environment is highly uncertain.<br>3. Technological developments are highly unpredictable.<br>4. Technologically, our industry is a very complex environment. |
| Sustainable Competitive Advantage | 1. We make the business more competitive.<br>2. We establish new markets.<br>3. We ensure the long-term viability of the business.<br>4. We achieve better utilization of resources.<br>5. The degree of leverage sales of other products and services.<br>6. We bring new clients to the business.<br>7. We retaining existing customers. |
| Market Performance | Indicate your firm's performance in the last year comparing to major competitors.<br>1. Sales volume growth.<br>2. Profit margin growth.<br>3. Market share growth.<br>4. Overall competitive position. |

**Table A2.** Collinearity check table.

| Model | | Estimate | S.E | T-value | Tolerance | VIF | $R^2$ |
|---|---|---|---|---|---|---|---|
| M1 | (constant) | 0.846 | 0.163 | 5.177 | | | 0.641 |
| | MO | 0.799 | 0.030 | 26.947 | 1.000 | 1.000 | |
| M2 | (constant) | 0.430 | 0.168 | 2.553 | | | 0.674 |
| | MO | 0.710 | 0.031 | 22.552 | 0.807 | 1.239 | |
| | MRI | 0.197 | 0.030 | 6.468 | 0.807 | 1.239 | |
| M3 | (constant) | 0.372 | 0.180 | 2.071 | | | 0.675 |
| | MO | 0.705 | 0.032 | 22.178 | 0.790 | 1.265 | |
| | MRI | 0.189 | 0.031 | 6.015 | 0.754 | 1.325 | |
| | MU | 0.025 | 0.027 | 0.932 | 0.866 | 1.155 | |
| M4 | (constant) | 0.361 | 0.179 | 2.014 | | | 0.677 |
| | MO | 0.708 | 0.032 | 22.270 | 0.788 | 1.269 | |
| | MRI | 0.189 | 0.031 | 6.006 | 0.754 | 1.326 | |
| | MU | 0.022 | 0.027 | 0.812 | 0.861 | 1.161 | |
| | MRI*MU | 0.043 | 0.027 | 1.582 | 0.992 | 1.008 | |
| N1 | (constant) | 2.883 | 0.212 | 13.607 | | | 0.280 |
| | MO | 0.484 | 0.038 | 12.575 | 1.000 | 1.000 | |
| N2 | (constant) | 2.120 | 0.212 | 9.994 | | | 0.398 |
| | MO | 0.352 | 0.038 | 9.212 | 0.850 | 1.176 | |
| | TRI | 0.311 | 0.035 | 8.915 | 0.850 | 1.176 | |
| N3 | (constant) | 2.044 | 0.224 | 9.116 | | | 0.399 |
| | MO | 0.342 | 0.039 | 8.717 | 0.804 | 1.244 | |
| | TRI | 0.306 | 0.035 | 8.686 | 0.834 | 1.199 | |
| | TU | 0.035 | 0.034 | 1.043 | 0.892 | 1.121 | |
| N4 | (constant) | 2.068 | 0.222 | 9.305 | | | 0.412 |
| | MO | 0.339 | 0.039 | 8.704 | 0.803 | 1.245 | |
| | TRI | 0.304 | 0.035 | 8.707 | 0.833 | 1.200 | |
| | TU | 0.042 | 0.033 | 1.247 | 0.888 | 1.126 | |
| | TRI*TU | −0.098 | 0.033 | −2.954 | 0.995 | 1.005 | |

Note. Dependent variable for M1 ~ M4 is MC, and for N1 ~ N4 is IC, S.E = standard error, VIF = variance inflation factor.

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
