# Peer review of "Determinants of a Firm’s Sustainable Competitive Advantages: Focused on Korean Small Enterprises"

_sustainability, doi:10.3390/su13010346_

Round 1

Reviewer 1 Report

My general comments are regarding with the link between the topic presented in the paper and sustainability. You mentioned several times sustainable advantage but it is not enough to establish a link with the economic component of sustainability. Therefore, more evidence and/or arguments should be offered to potential readers of your article in order to convince them that it deals with sustainability.

Specific comments:

line 328 - was kind of conference? specific firms where invited?

line 327 - what sampling procedure was implemented? was it a random or non-random sampling procedure?

line 328 - what kind of surveys? face to face? online?

line 338-339 - why did you chose those specific goodness of fit indicators? why not PClose, or SRMR, or p-value for the model?

in table 2 - for reliability you present both Cronbach`s alpha and I assume composite reliability (C/R) - why both?

Also for discriminant validity you should combine table 1 with AVE from table 2, because it will be easier for potential readers to asses at a glance the constructs discriminant validity.

Author Response

Dear anonymous reviewers

Please accept our article (sustainability-1057467), “Determinants of Firms’ Sustainable Competitive Advantage: Focused on Korean Small Enterprises” for continued consideration for publication in the Sustainability Journal. We have attempted to address each concern as noted by the review team. Consistent with this, we have uploaded our responses to each reviewer comment. Please see the attachment. The review team was very specific and helpful in their comments, and we believe our revised manuscript is much improved from our original submission.

Primary changes we have made are to add the detailed description of research method and provide additional statistical indices. We also have attempted to enhance the flow and readability of the paper. We are even more confident that our research now makes a contribution to existing marketing literature, especially in relation to the continued lack of empirical research on competitive advantage of small firms.

Thank you for your continued consideration of this manuscript. My co-author and I thank the editorial review team for the very specific guidance given in relation to our initial submission. We look forward to the continued review process, and welcome future feedback.

Sincerely,

Jaewon Yoo, Soongsil University

Sora Lee, Soongsil University

Reviewer 2 Report

In my view the literature review is sufficient but needs to be clearly tied to the empirical part. It is not evident were did the conceptual model came from.

The empirical part needs further attention from the authors, namely concerning that:

a) Environmental facts such as turbulence should appear as controls as they are exogenous factors, and it seems to me that they should be included in a different way.

b) Sampling procedure section needs reformulation as this section relates more to the respondent sample rather than the data collection method. As it is a primary data source, more details should be given to underline reliability (e.g. period of response, method of data collection....)

c) Additionally, more details about the questions that constitute the survey and the proxy construction are required as the magnitude of the coefficients cannot be fully understood as the reader does not understand if the survey relies on likert scales, perceptions, if the respondents' organisation is the same .....

d) Basic descriptives are missing to understand the proxies in use and their variability. Asymmetry measures such as skewness and Kurtosis will provide a clearer picture about potential outliers. 

e) The correlation table presents high levels for some variables - e.g. turbulence(s). These results make us suspicious about potential multicollinearity. I believe that  VIF tests should be run to appraise the independence levels among variables to verify the absence of over-inflated coefficients.

f) Running an OLS estimation to provide a robustness check would help generalization and bring  extra reliability to the model construction.

g) Moreover, when some effects appear as statistically insignificant, and forced by te parsimonia criterium, another model should be run removing the invalid predictors, in doing so some of the complexity will be removed.

Author Response

(The authors gave the same response as above.)
